# Comparing Selection Criteria to Select Grapevine Clones by Water Use Efficiency

**Andreu Mairata** [1,*] , **Ignacio Tortosa** [2] , **Cyril Douthe** [2] , **José Mariano Escalona** [2] , **Alicia Pou** [1] **and Hipólito Medrano** [2]

1   Departamento de Viticultura, Instituto de Ciencias de la Vid y del Vino (Gobierno de La Rioja, Universidad de La Rioja, CSIC), Finca La Grajera, Ctra. De Burgos Km 6, 26007 Logroño, Spain

2   INAGEA, Department of Biology, Universitat de les Illes Balears, cta. de Valldemossa Km 7.5, 07122 Palma de Mallorca, Spain

*   Correspondence: andreumairata112@gmail.com

**Abstract:** The current climate change is forcing growth-adapted genotypes with a higher water use efficiency (WUE). However, the evaluation of WUE is being made by different direct and indirect parameters such as the instantaneous leaf WUE ($WUE_i$) and isotopic discrimination of carbon ($\delta^{13}C$) content of fruits. In the present work, WUE has been evaluated in these two ways in a wide collection of grapevine genotypes, including Tempranillo and Garnacha clones, and Tempranillo on different rootstocks (T-rootstocks). A total of 70 genotypes have been analysed in four experimental fields over two years. The parameters used to measure WUE were the bunch biomass isotopic discrimination ($\delta^{13}C$) and the intrinsic WUE ($WUE_i$), defined as the ratio between net $CO_2$ assimilation and stomatal conductance. The genotypes with the highest and lowest WUE were identified, differences between them being found to be of more than 10%. Generally, the two parameters showed coincidences in the clones with the highest and lowest WUE, suggesting that both are valuable tools to classify genotypes by their WUE in grapevine breeding programs. However, $\delta^{13}C$ seemed to be a better indicator for determining WUE because it represents the integration over the synthesis time of the sample analysed (mainly sugars from ripening grapes), which coincides with the driest period for the crop. Moreover, the $WUE_i$ is a variable parameter in the plant and it is more dependent on the environmental conditions. The present work suggests that carbon isotopic discrimination could be an interesting parameter for the clonal selection criteria in grapevines by WUE. The main reasons were its better discrimination between clones, the fact that sampling is less time-consuming and easier to do than $WUE_i$, and that the samples can be stored for late determinations, increasing the number of samples that can be analysed.

**Keywords:** vitis; WUE; clonal selection; carbon isotope discrimination; photosynthesis; stomatal conductance; vid; breeding

## 1. Introduction

Grapevine is a traditional Mediterranean crop with a long history that completes its biological cycle during the driest and warmest months of the year. Vine cultivation is mainly located in semi-arid areas with an irrigation water contribution that implies the over-exploitation of available water [1,2]. Furthermore, climate change is causing more frequent and longer droughts and heatwaves combined with increasingly unpredictable torrential rainfall that reduces the actual soil available for the vines [3,4]. These grounds lead to troubling situations of economic and environmental conflict.

Spain is the country with the largest viticulture area in the world and is the third-biggest wine producer [5], predominantly in a Mediterranean climate where the irrigated vineyard area was 41.5% in 2021, 0.3% higher than in the previous year [6]. Consequently, current data and future predictions point to the important need to optimise irrigation water use to improve environmental sustainability and the economic balance of the crop.

Water use efficiency (WUE) in grapevine is a major topic in applied and fundamental research [7]. The research for drought-adapted cultivars and clones will become an indispensable requirement in semi-arid conditions. Previous work demonstrated the variability of WUE between cultivars and clones [8–13].

The favourable results in classic genetic selection, the existence of a very wide diversity of cultivated grapevine varieties [14,15] and the continuous progress in genomics [16] offer the genus vitis a wider genetic range to adapt grapevines to situations of increased water stress [17]. This background, coupled with continuous technological progress, offers the necessary conditions to find more drought-adapted grapevine genotypes.

Nowadays, the application of genomic and genetic engineering tools makes it very attractive for grape breeding due to the long time needed with traditional methods [18]. The utilization of molecular markers can easily identify quantitative trait loci (QTL) that affect traits of interest to accelerate the introduction in host plants using the backcrossing method [19]. The breeding method is assisted by molecular markers. Genetic engineering could make it possible to obtain new varieties/clones with, for example, high yield, disease resistance, different sugar content, early maturity, or drought tolerance [20]. Until now, very little progress has been seen in new commercial varieties.

In recent decades, the main selection programs developed were focused on clonal selection inside the more commercial varieties because of the legal frameworks of wine protection. Breeding new varieties would require a long administrative process and acceptance by regulatory boards and consumers. In contrast, the selection of clones within an authorised variety was immediately accepted [14].

One of the problems in the selection of genotypes by WUE is how to estimate this parameter. Conceptually, WUE reflects the balance between carbon gains and water loss. This balance can be measured at different levels from leaf instantaneous gas fluxes to plant production [21]. At the leaf level, the ratios between $CO_2$ assimilated ($A_N$) and transpiration (E) or stomatal conductance ($g_s$) determine the WUE of the plant. "Intrinsic water use efficiency" is determined by factors that the plant can control ($A_N/g_s$, $WUE_i$), less influenced by environmental conditions than the "instantaneous water use efficiency" ($A_N/E$, $WUE_{inst}$) [22].

These leaf determinations should be taken as representative of the water efficiency over the plant cycle. It is a selection criterion with a clear physiological basis, even though the daily and seasonal measurements of $WUE_i$ are "instantaneous". To overcome these limitations, biomass determination of stable carbon isotope abundance, in particular the $^{13}C$ ratio, was proposed as a reliable indicator of WUE [23–25].

Photosynthetic processes discriminate between the $^{12}C$ and $^{13}C$ isotopes due to their different diffusion between the atmosphere and chloroplasts. This discrimination against $^{13}C$ ($\delta^{13}C$) also occurs in the ribulose biphosphate carboxylase/oxidase (RuBisCo) reaction catalysed by the Rubisco enzyme and is attenuated when the $CO_2$ concentration in chloroplasts decreases due to stomatal closure. In consequence, the differential proportion of carbon isotopes in plant dry matter results in an integrative estimate of the relationship between photosynthetic rate and stomatal aperture ($WUE_i$) throughout the synthesis period of the analysed biomass [25].

Tempranillo and Garnacha are among the most widely cultivated varieties in Spain [14]. In addition, the use of drought-tolerant rootstocks in grapevine helps minimise the effect of water stress through improved water uptake and transport [26], also controlling plant transpiration through chemical response [27] and hydraulic signalling [28].

Measurements of $\delta^{13}C$ and $WUE_i$ have been used in previous work as indicators of WUE in the grapevine [12,13,23,29]. In this context, the objectives of this work were: (i) Analyse the variability of WUE between clones of the Garnacha and Tempranillo cultivar and Tempranillo on different rootstocks (T-rootstocks), (ii) Evaluate the discrimination capacity of $WUE_i$ and $^{13}C$ isotopic ratio in two years of field-growing vines data and, (iii) Compare both parameters as operative selection criteria by their interest in grapevine clone breeding.

## 2. Materials and Methods

### 2.1. Experimental Sites and Plant Material

The experiment was carried out in four experimental plots: two located in Logroño (La Rioja, Spain), one in Haro (La Rioja, Spain) and the last one in Miranda de Arga (Navarra, Spain). In total, 58 clones of two cultivars (Garnacha and Tempranillo) and 12 genotypes of rootstocks on Tempranillo (T-rootstocks) were measured over two years (2015 and 2018). Clone groups were randomly distributed in each experimental plot.

In each field, leaf gas exchange measurements ($WUE_i$) were realised in August and berry samples ($\delta^{13}C$) were collected at maturity in September and October. The environmental conditions of the climatic stations closest to the experimental fields were described in the two years of study (Table 1). Data were collected and averaged by month from 1 April to 31 October. Growing degree days (in $°C\ day^{-1}$) were calculated as daily $T_{mean} - T_{base}$ (only positive values, $T_{base} = 10\ °C$), and reference evapotranspiration (ET0) was calculated using the Penman–Monteith method [30,31].

### 2.2. Leaf Gas Exchange Measurements

Instantaneous leaf gas exchange measurements were done using an open infrared gas analyser system (Li-6400xt; Li-Cor, Inc. Lincoln, NE, USA). Leaf net photosynthesis ($A_N$) and stomatal conductance ($g_s$) were measured in a fully exposed mature leaf (one measure per plant and 4–6 plants per clone). The $CO_2$ concentration reference was 400 μmol $CO_2\ mol^{-1}$ air with a flow rate of 350 μmol (air) $min^{-1}$. All the measurements were always taken above the 1500 μmol $m^2\ s^{-1}$ active photosynthetic radiation (PAR) between 10:00 and 13:00 (local time) using a 6 $cm^2$ chamber [13]. Intrinsic water use efficiency ($WUE_i$) was calculated as the $A_N$ and $g_s$ ratio.

### 2.3. Carbon Isotope Ratios

The carbon isotope ratio ($\delta^{13}C$) was determined from samples of 30 berries/plants collected at harvest, at the same plants measured for $WUE_i$ (4–6 plants per clone). Berry samples were oven-dried (taking the seed out) and $\delta^{13}C$ was analysed in $2 \pm 0.1$ mg aliquots of berry powder samples (Thermo Flash EA 1112 Series) [23]. Determinations of $\delta^{13}C$ were carried out using an Elemental analyser (NC2500, Carlo Erba Reagents) coupled to an isotope ratio mass spectrometer (Thermoquest Delta Plus, ThermoFinnigan). The carbon isotope ratio was expressed as $\delta^{13}C = [(R_s - R_b)/R_b] \times 1000$ [25], where $R_s$ is the ratio $^{13}C/^{12}C$ of the sample. $R_b$ is the $^{13}C/^{12}C$ of the PDB (PeeDee Belemnite) standard (0.0112372) and was measured every seven samples.

### 2.4. Statistical Analysis

Every cultivar and plot was analysed independently due to their differences in climate, soil, crop management, vine characteristics, etc. Two-way analysis of variance (ANOVA) was used to evaluate the effects of the factors and their interactions on all the variables measured and calculated (Table 2). Then, the $WUE_i$–$\delta^{13}C$ regressions obtained in each group were demonstrated. Seven separated (one per group) one-way ANOVAs were performed to check where the parameters were significant. Distribution and homoscedasticity were analysed using the Shapiro–Wilk test and Levene's statistic. When differences were found, a post-hoc test (Duncan) was applied to determine which genotypes were different and estimate a ranking [32]. Data analyses were performed with SPSS 22.0 (IBM Corp., Armonk, NY, USA). Any differences were accepted with a *p*-value > 0.05.

**Table 1.** Climatic conditions of the three experimental sites. The values are the average of the maximum (Tmax) and minimum temperature (Tmin) and the sum of the cumulative precipitation (P), the reference evapotranspiration (ET0) and the growing degree days (GDD) accumulative by months from April to October in 2015 and 2018 [30,31].

| | Year | T max (°C) | T min (°C) | P (L m$^{-2}$) | ET0 (mm Month$^{-1}$) | GDD (°C Month$^{-1}$) |
|---|---|---|---|---|---|---|
| | | | | Roda | | |
| April | | 18.1 | 6.9 | 16.2 | 113.1 | 72.7 |
| May | | 21.8 | 10.0 | 7.2 | 144.6 | 169.7 |
| June | | 26.6 | 12.3 | 63.8 | 158.6 | 268.9 |
| July | 2015 | 30.6 | 15.2 | 10.7 | 190.1 | 370.6 |
| August | | 28.5 | 14.1 | 37.1 | 167.8 | 339.3 |
| September | | 22.1 | 10.5 | 26.8 | 98.8 | 170.1 |
| October | | 18.1 | 8.6 | 58.8 | 65.9 | 96.9 |
| | | 23.7 | 11.1 | 220.6 | 938.9 | 1488.2 |
| April | | 17.1 | 7.6 | 97.7 | 97.8 | 76.0 |
| May | | 19.9 | 8.5 | 51.3 | 111.4 | 120.2 |
| June | | 25.0 | 12.8 | 33.3 | 134.6 | 249.6 |
| July | 2018 | 28.4 | 15.6 | 78.9 | 156.3 | 347.7 |
| August | | 29.6 | 14.7 | 0.0 | 157.2 | 356.4 |
| September | | 27.4 | 13.6 | 41.4 | 113.8 | 287.3 |
| October | | 19.0 | 8.3 | 66.0 | 65.0 | 115.3 |
| | | 23.8 | 11.6 | 368.6 | 836.1 | 1552.5 |
| | | | | La Grajera–Vitis Provedo | | |
| April | | 18.7 | 7.3 | 21.0 | 105.0 | 81.7 |
| May | | 22.6 | 10.8 | 2.6 | 142.5 | 196.0 |
| June | | 27.7 | 14.1 | 42.8 | 170.7 | 307.5 |
| July | 2015 | 31.5 | 17.0 | 34.9 | 197.8 | 410.1 |
| August | | 29.0 | 15.3 | 19.1 | 163.9 | 364.0 |
| September | | 23.0 | 11.6 | 13.3 | 100.7 | 204.6 |
| October | | 18.5 | 9.1 | 33.2 | 62.9 | 111.4 |
| | | 24.4 | 12.2 | 166.9 | 943.5 | 1675.3 |
| April | | 17.3 | 7.5 | 86.1 | 90.6 | 76.7 |
| May | | 19.9 | 9.8 | 65.6 | 112.7 | 138.4 |
| June | | 25.2 | 13.9 | 39.3 | 137.4 | 272.0 |
| July | 2018 | 28.9 | 16.7 | 117.6 | 168.7 | 371.3 |
| August | | 30.0 | 16.4 | 0.0 | 167.2 | 386.1 |
| September | | 27.3 | 14.5 | 45.4 | 113.1 | 302.0 |
| October | | 19.5 | 9.3 | 28.3 | 70.7 | 132.4 |
| | | 24.0 | 12.6 | 382.3 | 860.4 | 1678.9 |
| | | | | Vitis Navarra | | |
| April | | 19.8 | 6.0 | 11.9 | 101.1 | 85.3 |
| May | | 23.5 | 10.2 | 2.4 | 148.9 | 206.7 |
| June | | 28.3 | 13.4 | 70.7 | 165.9 | 317.7 |
| July | 2015 | 30.9 | 16.0 | 0.3 | 184.2 | 395.9 |
| August | | 28.8 | 14.2 | 10.3 | 142.7 | 350.4 |
| September | | 23.5 | 10.4 | 12.9 | 95.2 | 202.6 |
| October | | 19.9 | 8.5 | 24.1 | 57.0 | 116.8 |
| | | 24.9 | 11.3 | 132.6 | 895.0 | 1675.4 |
| April | | 18.3 | 6.5 | 61.3 | 93.1 | 80.3 |
| May | | 21.0 | 9.0 | 29.2 | 119.4 | 149.3 |
| June | | 25.8 | 12.9 | 48.6 | 145.8 | 275.9 |
| July | 2018 | 30.2 | 15.7 | 56.3 | 174.8 | 386.9 |
| August | | 29.8 | 14.9 | 6.3 | 159.4 | 372.4 |
| September | | 28.0 | 12.9 | 16.2 | 113.3 | 300.4 |
| October | | 20.7 | 8.4 | 16.6 | 72.6 | 141.1 |
| | | 24.8 | 11.5 | 234.5 | 878.4 | 1706.3 |

**Table 2.** $g_s$ and intrinsic water use efficiency (WUE$_i$) average and their standard deviations in Tempranillo, Garnacha and rootstock cultivars in the different fields and years analysed.

| | | 2015 | | 2018 | |
|---|---|---|---|---|---|
| | | $g_s$ (mol H$_2$O m$^{-2}$ s$^{-1}$) | WUE$_i$ (mmol CO$_2$ mol$^{-1}$ H$_2$O) | $g_s$ (mol H$_2$O m$^{-2}$ s$^{-1}$) | WUE$_i$ (mmol CO$_2$ mol$^{-1}$ H$_2$O) |
| Tempranillo | Roda | | | 0.311 ± 0.108 a | 57.8 ± 15.5 c |
| | La Grajera | 0.097 ± 0.045 c | 118.9 ± 23.6 a | 0.098 ± 0.054 c | 90.5 ± 14.1 a |
| | Vitis Provedo | 0.198 ± 0.072 a | 76.7 ± 17 b | | |
| | Vitis Navarra | 0.164 ± 0.063 b | 76.5 ± 16.5 b | | |
| Garnacha | Vitis Navarra | | | 0.236 ± 0.048 b | 70.6 ± 11.3 b |
| T-Rootstock | Vitis Navarra | | | 0.233 ± 0.095 b | 64.6 ± 13.9 bc |
| | General | 0.126 ± 0.069 B | 103.2 ± 29.1 A | 0.236 ± 0.104 A | 68 ± 16.8 B |
| | G$_s$: Two-way ANOVA: Year **, Field ***, Year × Field ** EUA$_i$: Two-way ANOVA: Year ***, Field ***, Year × Field *** | | | | |

Different lower case letters indicate a difference between groups of the same year with Duncan's test ($p < 0.05$). *** $p$-value < 0.001; ** $p$-value < 0.01.

## 3. Results

### 3.1. Comparison of Fields and Year Effect

The climatic data of the areas of the experimental plots were analysed: three fields located in the region of La Rioja and the other in the region of Navarra, both located in the north of Spain (Table 1). La Grajera and Vitis Provedo fields were located in Logroño (La Rioja), and the third one in Haro (West of La Rioja), near Roda's winery. Vitis Navarra was located near Larraga, Navarra. All fields were characterised by a Mediterranean climate, with a warm and low rainfall in summer. Roda's plot is characterised by a less warm summer, with a 10% lower accumulation of growing degrees. Generally, Vitis Navarra had less precipitation and drier climate conditions. In 2018, there was a precipitation increase of 129% in La Grajera and Vitis Provedo, 67% in Roda and 77% in Navarra fields (compared with 2015). Nevertheless, GDD, ET0 and temperatures remained very stable between the two years.

Stomatal conductance is determined by plant water status and, at the time, determines WUE [33]. For this reason, the water status was estimated as $g_s$ for all the plots and cultivars (Table 2). The La Grajera farm showed higher water stress ($g_s < 0.1$ mol H$_2$O m$^{-2}$ s$^{-1}$) in both years. Interestingly, the difference in rainfall does not determine the water status of the plants between plots in the same year. Roda's field (2018) showed a higher stomatal conductance, reaching values close to 0.5 mol H$_2$O m$^{-2}$ s$^{-1}$.

Significant differences in $g_s$ and WUE$_i$ were observed between fields, cultivars and years. Comparing the average water status between the two years, there was an increase (2018 reached 2015) in stomatal conductance (+87%) and, therefore, a significant decrease in WUE$_i$ (−59%).

Even though there was a large range of $g_s$ in the experimental fields, a good correlation ($R^2$: 0.7686) was founded between ln WUE$_i$ and $g_s$ values in all groups, plots and years analysed (Figure 1).

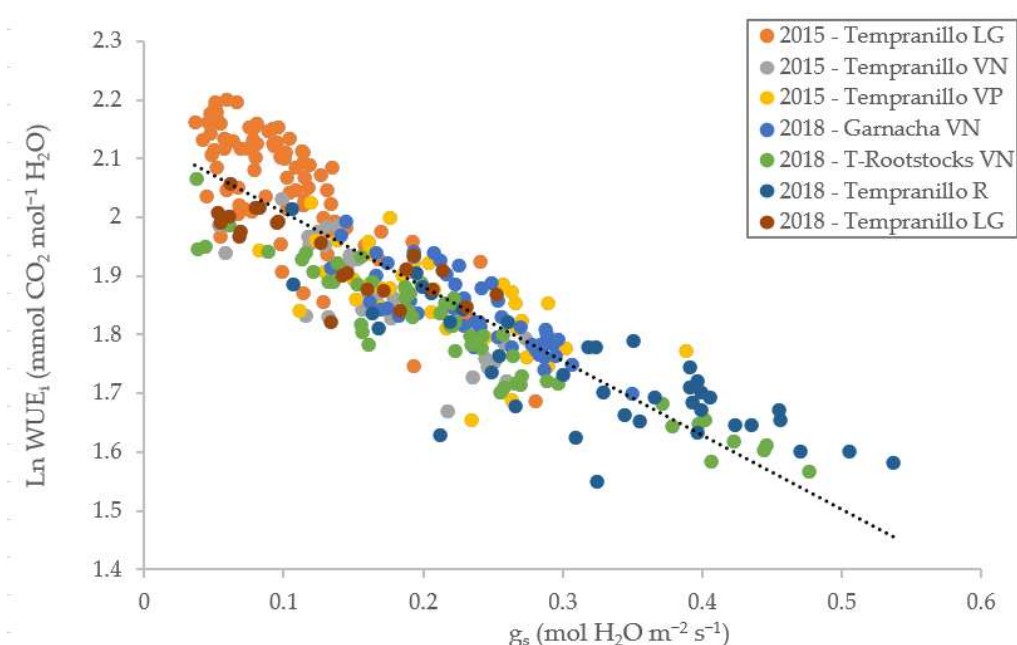

**Figure 1.** Linear regression between the natural logarithm intrinsic water use efficiency ($WUE_i$, $A_N/g_s$) and stomatal conductance ($g_s$) representing the clonal groupings analysed (LG: La Grajera; VN: Vitis Navarra; VP: Vitis Provedo; R: Roda).

*3.2. Genotypic Characterisation of WUE*

Due to the high variability in water status between plots and years (Table 2), an independent analysis was carried out for each field, year and cultivar. Table 3 shows the average $WUE_i$ data of the clones analysed by year, cultivar and plot. Letters represent significant differences between genotypes ($p$-value < 0.05). A total of seven independent analyses were carried out. Only Tempranillo's cultivar in Vitis Provedo (2015) showed non-significant differences between clones in $WUE_i$. For the $WUE_i$, a maximum value of 143.1 mmol $CO_2$ mol$^{-1}$ $H_2O$ and a minimum of 40.8 mmol $CO_2$ mol$^{-1}$ $H_2O$ were obtained at 1048 clone (Tempranillo, La Grajera) in 2015 and 137 clone (Tempranillo, Roda) in 2018, respectively. Great variability was observed between clones of the same group, for example, between the genotype 140RU and RG9 (T-rootstocks, 2018), where the percentage increase was 91.55%.

The same statistical analysis was performed with the $\delta^{13}C$ data (Table 4). Significant differences in $^{13}C$ content between clones were observed in all groups. The mean values of $^{13}C$ have a range of 8 ‰, including values between −21.5 ‰ (clone RG8, T-Rootstocks, 2018) and −29.4 ‰ (clone 807, Tempranillo, La Grajera, 2015).

Integrating both parameters, the high efficiency of Tempranillo clones 807 was clearly shown (La Grajera, 2015), as well as that of VN32 (Vitis Navarra, 2015), 1048 (La Grajera, 2018) and six (Roda, 2018), that of the Garnacha clone ENTAV 136 (Vitis Navarra, 2018) and the T-rootstock clone RG2 (Vitis Navarra, 2018). In addition, some genotypes stand out for their low WUE, including Tempranillo clones 1084 (La Grajera, 2015) and 137 (Roda, 2018), the Garnacha clone EV15 (Vitis Navarra, 2018) and the T-rootstock clone RG8 (Vitis Navarra, 2018).

**Table 3.** Mean values and standard deviations of $WUE_i$ ($A_N/g_s$) in the genotypes and clones studied.

| | | | | | |
|---|---|---|---|---|---|
| **2015** | | | | | |
| **Tempranillo** | | | | | |
| **La Grajera** | | **Vitis Navarra** | | **Vitis Provedo** | |
| 86 | 110.7 ± 19.8 bcdef | VN1 | 74.3 ± 9.5 b | RJ43 | 69.4 ± 6.8 n.s. |
| 232 | 123.7 ± 18.8 abcd | VN31 | 77.5 ± 8.5 b | RJ78 | 83.3 ± 27.4 n.s. |
| 260 | 131 ± 2.6 ab | VN32 | 97 ± 6.5 a | VP11 | 65.6 ± 9.3 n.s. |
| 280 | 121.5 ± 15.2 abcd | VN33 | 64.9 ± 12.7 b | VP24 | 83.9 ± 9.8 n.s. |
| 518 | 132.4 ± 9.3 ab | VN42 | 79.6 ± 22.2 ab | VP25 | 87.3 ± 13.6 n.s. |
| 560 | 124.7 ± 34 abcd | VN69 | 65.6 ± 16.3 b | VP28 | 66.7 ± 10.2 n.s. |
| 807 | 129.4 ± 26.8 ab | | | VP8 | 80.9 ± 23.6 n.s. |
| 814 | 136.3 ± 10.6 ab | | | | |
| 843 | 129 ± 5.5 abc | | | | |
| 1041 | 90.1 ± 24.8 ef | | | | |
| 1048 | 143.1 ± 8.9 a | | | | |
| 1084 | 81.7 ± 18.8 f | | | | |
| 1089 | 96.6 ± 25 def | | | | |
| RJ26 | 132.9 ± 22.8 ab | | | | |
| RJ43 | 113.1 ± 15.4 abcde | | | | |
| RJ51 | 99.3 ± 18.5 cdef | | | | |
| RJ78 | 126.5 ± 14.5 abcd | | | | |
| RJ79 | 123.1 ± 18 abcd | | | | |
| **2018** | | | | | |
| **Garnacha** | | **T -Rootstock** | | **Tempranillo** | |
| **Vitis Navarra** | | **Vitis Navarra** | | **La Grajera** | **Roda** |
| ARA-2 | 79.2 ± 8.2 ab | 1103P | 74.7 ± 5 ab | 232 | 93.7 ± 14.2 ab | 6 | 79.6 ± 5 a |
| ARA-24 | 57 ± 4.7 e | 110R | 66.8 ± 6.2 abcd | 1048 | 105.1 ± 6.4 a | 108 | 49.4 ± 9 c |
| ARA-4 | 81.6 ± 7.9 ab | 140RU | 41.4 ± 2.8 e | 1052 | 101.1 ± 3.3 ab | 137 | 40.8 ± 3.4 c |
| ENTAV 136 | 88.6 ± 5.7 a | 420A | 64.1 ± 8.2 bcd | 1078 | 88 ± 10 abc | 156 | 47.9 ± 4.8 c |
| ENTAV 141 | 73.5 ± 10.9 bcd | RG2 | 72.6 ± 8 ab | 1084 | 72.8 ± 6.4 c | 166 | 48.2 ± 1.1 c |
| EV11 | 67.3 ± 1.8 cde | RG3 | 55 ± 11.8 d | 1371 | 86.1 ± 14.2 bc | 178 | 77.2 ± 16.3 ab |
| EV13 | 65.5 ± 9.4 de | RG4 | 58.1 ± 6.1 cd | | | 203 | 66.4 ± 9.3 b |
| EV14 | 63.2 ± 5.6 e | RG6 | 71.6 ± 5.4 ab | | | 215 | 52.6 ± 4.6 c |
| EV15 | 63.1 ± 6.7 e | RG7 | 75.2 ± 9.9 ab | | | 243 | 46.2 ± 5.8 c |
| RJ21 | 61.1 ± 1.5 e | RG8 | 42.4 ± 4.9 e | | | 336 | 66.9 ± 7.1 b |
| VNQ | 75.7 ± 5.1 bc | RG9 | 79.3 ± 11.9 a | | | | |
| | | SO4 | 68.5 ± 11.8 abc | | | | |

Different letters indicate statistical differences within each group by Duncan test ($p < 0.05$).

An integrator value was obtained for the genotype in each group by adding the proportional distribution of the relative standard deviation of the values obtained in $WUE_i$ and $\delta^{13}C$. With this method, it was possible to define water efficiency for each clone according to the values obtained for both parameters. The clones were defined as very efficient (residual > 15%), efficient (15 to 5%), normal (5 to −5%), inefficient (−5 to −15%) or very inefficient (<−15%), depending on the values obtained of the calculated percentages. Of the 70 clones analysed, 14 showed to be very efficient and 21 to be very inefficient. 1048 genotype (Tempranillo) was defined as a very efficient genotype in both years. In the Tempranillo cultivar (largest number of clones analysed), very efficient genotypes in water use efficiency were 814 and 1048 (La Grajera, 2015), VN32 (Vitis Navarra, 2015), VP25 (Vitis Provedo, 2015), 1048 (La Grajera, 2018) and 6, 178, 203 and 336 (Roda, 2018). In contrast, genotypes defined as very inefficient were 1041, 1084, 1089 and RJ51 (La Grajera, 2015), VN33 and VN69 (Vitis Navarra, 2015), RJ43 and VP11 (Vitis Provedo, 2015), 1084 and 1371 (La Grajera, 2018) and 108, 137,156,166 and 243 (Roda, 2018). In the Garnacha cultivar (Vitis Navarra, 2018), the genotype defined as very efficient was ENTAV 136, and the very

inefficient ones were ARA-24, EV15 and RJ21. The T-rootstock genotypes (Vitis Navarra, 2018), clones defined as very efficient were 1103P, RG2, RG7 and RG9. In contrast, the very inefficient ones were 140RU, RG3 and RG8.

**Table 4.** Mean values and standard deviations of $^{13}C$ isotopic discrimination ($\delta^{13}C$ ‰) in the genotypes and clones studied.

| 2015 | | | | | |
|---|---|---|---|---|---|
| **Tempranillo** | | | | | |
| **La Grajera** | | **Vitis Navarra** | | **Vitis Provedo** | |
| 86 | $-22 \pm 0.8$ [abc] | VN1 | $-27.1 \pm 0.7$ [bc] | RJ43 | $-26.3 \pm 0.4$ [a] |
| 232 | $-21.7 \pm 0.2$ [a] | VN31 | $-27.8 \pm 0.4$ [c] | RJ78 | $-25.9 \pm 0.5$ [a] |
| 260 | $-22.4 \pm 0.4$ [abcd] | VN32 | $-25.3 \pm 0.7$ [a] | VP11 | $-26.6 \pm 0.4$ [a] |
| 280 | $-22.2 \pm 0.2$ [abcd] | VN33 | $-27.4 \pm 0.5$ [bc] | VP24 | $-24.4 \pm 0.6$ [c] |
| 518 | $-22.9 \pm 0.5$ [bcdef] | VN42 | $-27.3 \pm 0.8$ [bc] | VP25 | $-24.3 \pm 0.4$ [c] |
| 560 | $-21.8 \pm 0.9$ [ab] | VN69 | $-26.6 \pm 0.6$ [b] | VP28 | $-24 \pm 0.3$ [c] |
| 807 | $-21.5 \pm 0.2$ [a] | | | VP8 | $-25.2 \pm 0.6$ [b] |
| 814 | $-22.6 \pm 0.6$ [abcde] | | | | |
| 843 | $-22.4 \pm 0.4$ [abcd] | | | | |
| 1041 | $-24.8 \pm 0.5$ [gh] | | | | |
| 1048 | $-23 \pm 0.5$ [cdef] | | | | |
| 1084 | $-25.8 \pm 1.8$ [h] | | | | |
| 1089 | $-24.1 \pm 0.5$ [fg] | | | | |
| RJ26 | $-23.9 \pm 0.7$ [fg] | | | | |
| RJ43 | $-24 \pm 0.3$ [fg] | | | | |
| RJ51 | $-23.9 \pm 0.7$ [fg] | | | | |
| RJ78 | $-23.6 \pm 0.4$ [ef] | | | | |
| RJ79 | $-23.2 \pm 0.5$ [def] | | | | |
| 2018 | | | | | | | |
| **Garnacha** | | **T -Rootstock** | | **Tempranillo** | | | |
| **Vitis Navarra** | | **Vitis Navarra** | | **La Grajera** | | **Roda** | |
| ARA-2 | $-24.8 \pm 0.7$ [bc] | 1103P | $-26.3 \pm 0.3$ [b] | 232 | $-22.7 \pm 0.8$ [a] | 6 | $-24.9 \pm 0.8$ [ab] |
| ARA-24 | $-24.8 \pm 0.1$ [bc] | 110R | $-28 \pm 0.4$ [ef] | 1048 | $-23.2 \pm 0.1$ [a] | 108 | $-26.6 \pm 0.3$ [de] |
| ARA-4 | $-24.8 \pm 0.4$ [bc] | 140RU | $-28.5 \pm 0.5$ [f] | 1052 | $-23.2 \pm 0.8$ [a] | 137 | $-27.1 \pm 0.9$ [e] |
| ENTAV 136 | $-24.6 \pm 0.3$ [abc] | 420A | $-27.7 \pm 0.3$ [de] | 1078 | $-23.9 \pm 0.7$ [a] | 156 | $-26.6 \pm 0.4$ [de] |
| ENTAV 141 | $-27 \pm 0.6$ [e] | RG2 | $-25.4 \pm 0.7$ [a] | 1084 | $-24.1 \pm 0.6$ [a] | 166 | $-26.2 \pm 0.2$ [cde] |
| EV11 | $-24 \pm 0.7$ [a] | RG3 | $-27.4 \pm 0.4$ [cde] | 1371 | $-25.8 \pm 2.3$ [b] | 178 | $-24.9 \pm 0.7$ [ab] |
| EV13 | $-24.2 \pm 0.2$ [ab] | RG4 | $-27.1 \pm 0.9$ [bcd] | | | 203 | $-24.5 \pm 0.5$ [a] |
| EV14 | $-24.2 \pm 0.6$ [ab] | RG6 | $-26.7 \pm 0.4$ [bc] | | | 215 | $-26 \pm 0.8$ [cd] |
| EV15 | $-25.8 \pm 0.6$ [de] | RG7 | $-26.8 \pm 0.4$ [bc] | | | 243 | $-25.6 \pm 0.6$ [bc] |
| RJ21 | $-24.9 \pm 0.2$ [bc] | RG8 | $-29.4 \pm 0.3$ [g] | | | 336 | $-24.7 \pm 0.9$ [ab] |
| VNQ | $-25.2 \pm 0.3$ [cd] | RG9 | $-26.9 \pm 0.8$ [bcd] | | | | |
| | | SO4 | $-26.4 \pm 0.7$ [b] | | | | |

Different letters indicate statistical differences within each group by Duncan test ($p < 0.05$).

### 3.3. Ability of $\delta^{13}C$ and Leaf Gas Exchange Values to Measure WUE

The relationship between values of the two estimates of the WUE was analysed in Figure 2 which represents the relationship between $\delta^{13}C$ and $WUE_i$ values among the seven groups of genotypes analysed. A significant relationship (Pearson correlation of 0.699) was observed between both parameters (*p*-value < 0.05). Only in the Garnacha cultivar of Vitis Navarra (2018) was this relationship insignificant.

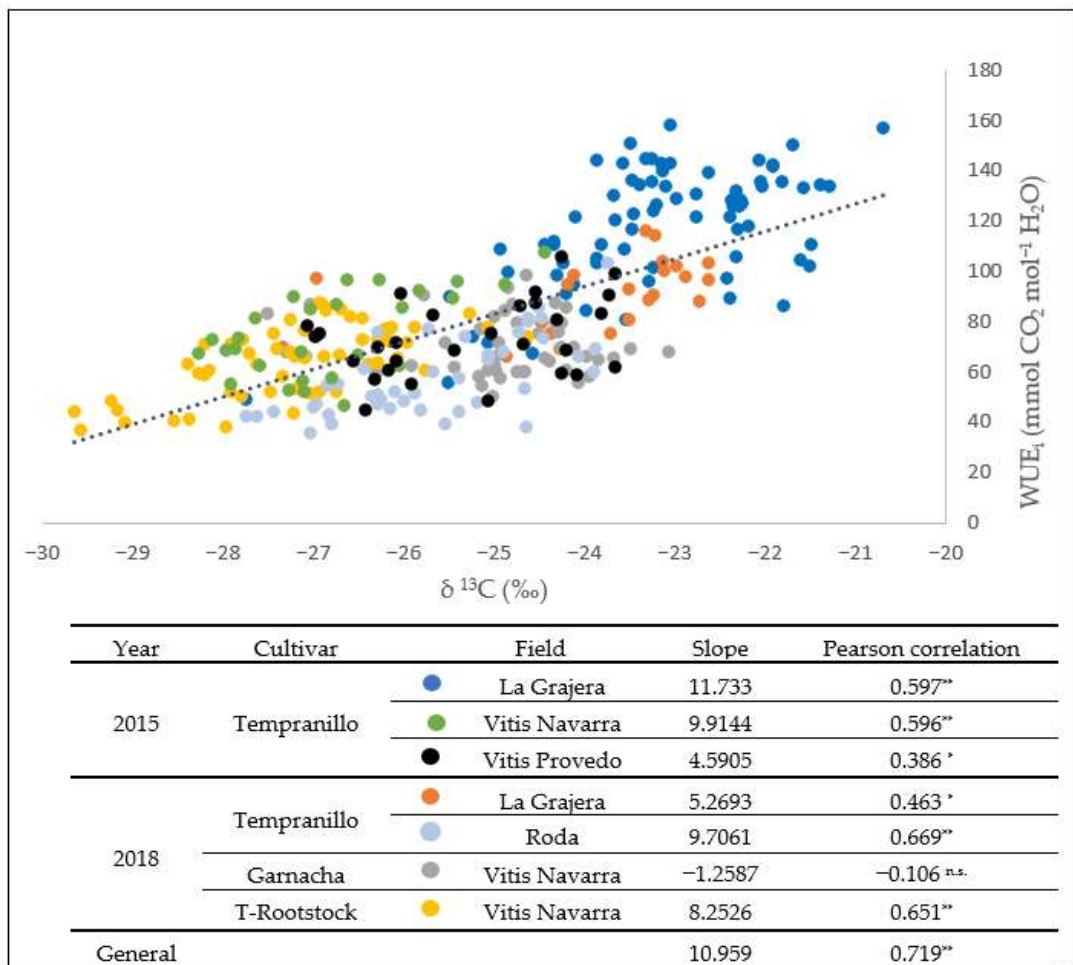

| Year | Cultivar | | Field | Slope | Pearson correlation |
|---|---|---|---|---|---|
| 2015 | Tempranillo | ● | La Grajera | 11.733 | 0.597** |
| | | ● | Vitis Navarra | 9.9144 | 0.596** |
| | | ● | Vitis Provedo | 4.5905 | 0.386* |
| 2018 | Tempranillo | ● | La Grajera | 5.2693 | 0.463* |
| | | ● | Roda | 9.7061 | 0.669** |
| | Garnacha | ● | Vitis Navarra | −1.2587 | −0.106 n.s. |
| | T-Rootstock | ● | Vitis Navarra | 8.2526 | 0.651** |
| General | | | | 10.959 | 0.719** |

**Figure 2.** Linear regression between carbon isotopic discrimination ($^{13}C$) and $WUE_i$ of the data set grouped in the clone sets analysed (LG: La Grajera; VN: Vitis Navarra; VP: Vitis Provedo; R: Roda). ** *p*-value < 0.01; * *p*-value < 0.05; n.s: not significant.

A good correlation was also shown for the values of the residual percentages between the $\delta^{13}C$ and $WUE_i$ (Figure 3). The $WUE_i$ percentages have a larger range of oscillation, reaching 37.5% compared to 11.4% for $\delta^{13}C$. Of the 70 genotypes analysed, 17 did not match the trend of the mean values of the residual percentages. Interestingly, 6 of the 11 clones of the Garnacha cultivar (Vitis Navarra, 2018) did not follow the general trend. However, there was a great general relationship (Pearson correlation of 0.585; *p*-value < 0.01) between the residual percentages of the parameters.

Once the genotypes were ranked by $WUE_i$ or $\delta^{13}C$, their relative position was quite coincident for both parameters. As shown in (Figure 4), each genotype value was correlated according to its value in both parameters ($WUE_i$ and $\delta^{13}C$). Interestingly, 83% of the genotypes were positioned in close to three positions in both parameters and 71% of genotypes in less than two positions. In general, there was a good correlation in the relative position of genotypes classified as best and worst according to the WUE.

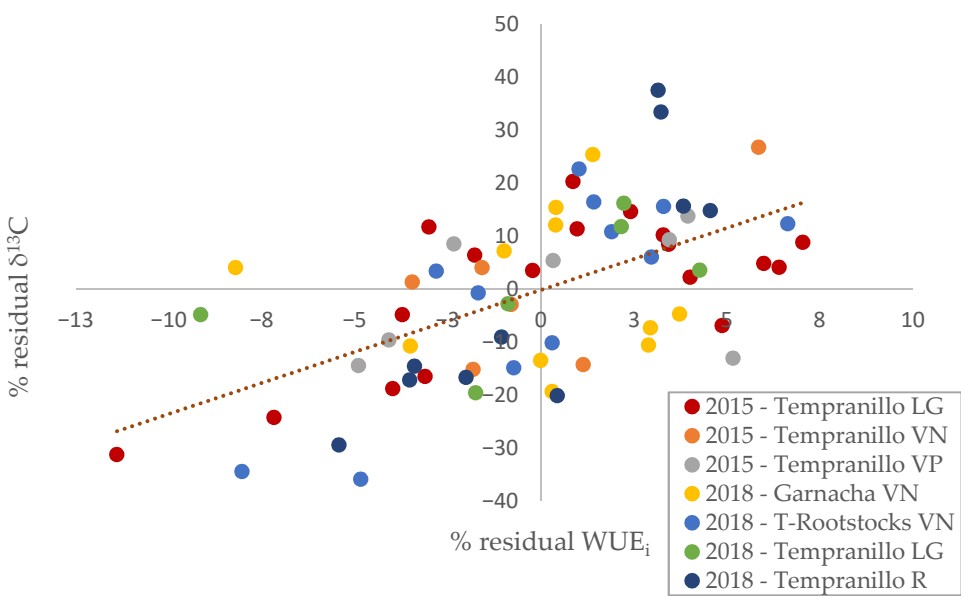

**Figure 3.** Residual percentages relationship of intrinsic water use efficiency (WUE$_i$) and carbon isotopic discrimination ($^{13}$C) in the clonal groupings analysed (LG: La Grajera; VN: Vitis Navarra; VP: Vitis Provedo; R: Roda).

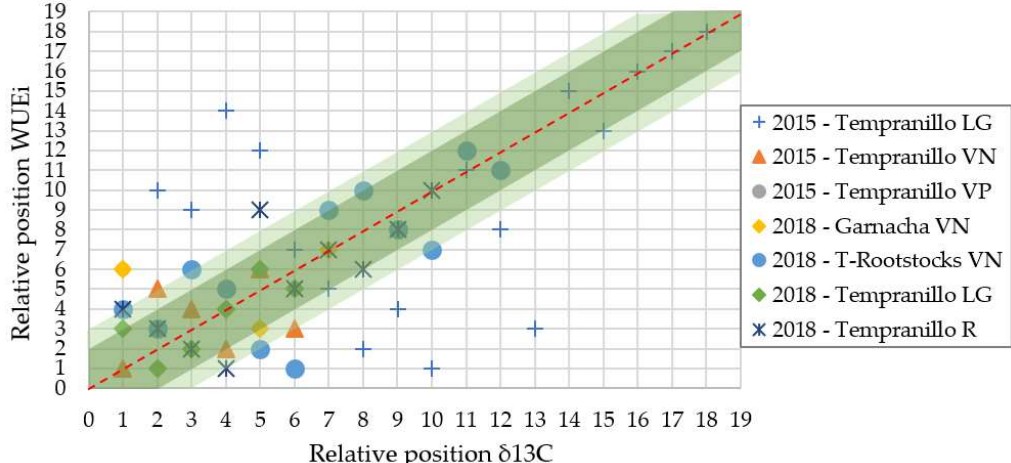

**Figure 4.** Relative position based on the residual percentage value of carbon isotopic discrimination ($^{13}$C) and intrinsic water use efficiency (WUE$_i$) in each group of clones and genotypes analysed (LG: La Grajera; VN: Vitis Navarra; VP: Vitis Provedo; R: Roda).

The correspondence between the $\delta^{13}$C and WUE$_i$ values was also analysed for the values of the genotypes defined according to their efficiency (Figure 5). The clones defined as efficient (formed by the very efficient and efficient genotypes) were located in areas with lower $^{13}$C discrimination and higher WUE$_i$ than the inefficient ones (formed by the very inefficient and inefficient genotypes). In general, in each subgroup, the $^{13}$C discrimination values showed a greater range of values for very similar WUE$_i$ values. This fact induces a better identification of better and poorer clones in WUE.

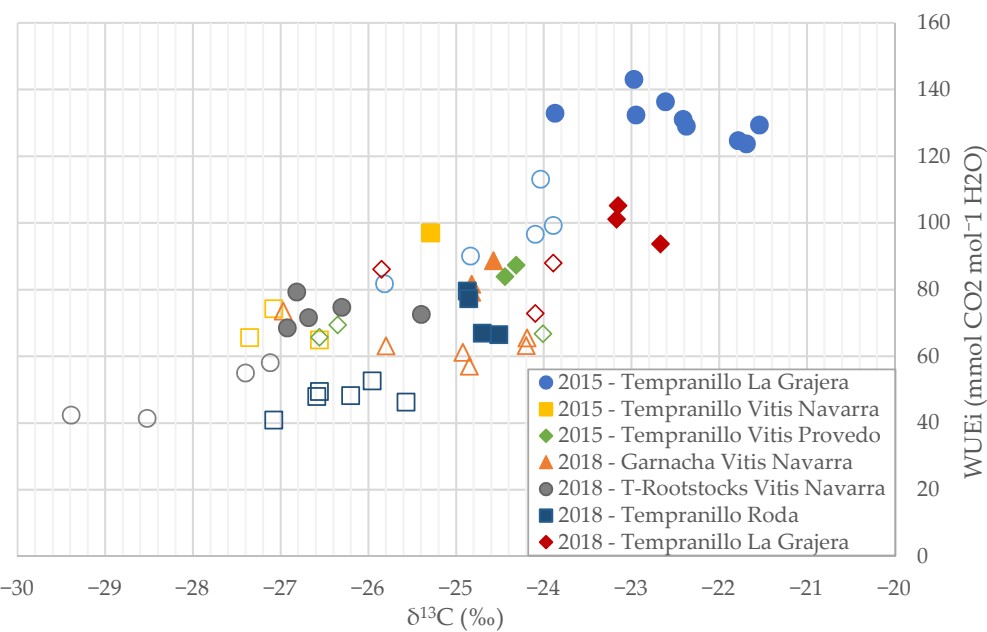

**Figure 5.** Linear regression between carbon isotopic discrimination ($^{13}$C) and WUE$_i$ of the analysed groups separating efficient and very efficient genotypes (filled mark) and very inefficient genotypes (empty mark).

## 4. Discussion

The genotype selection by WUE presents crucial limitations due to the difficulties of estimating the WUE of the whole plant. This difficulty has been reflected in other works with other grapevine cultivars and treatments [7,8,10,34,35].

Nevertheless, previous work demonstrated the existence of genetic variability in WUE determined as instantaneous values of WUE$_i$ between vine varieties and between clones of the Tempranillo cultivar [8,12,36]. The demonstration of this variability opens the way to initiate a breeding program to identify genotypes with higher (or lower) WUE. In this sense, recent works found differences in clones by WUE belonging to different years and locations [13,36].

The $\delta^{13}$C is a reputed parameter which enables to scale up from the water status of the plant [37,38]. At least conceptually, it shows the advantage of providing an overall WUE to be an integrator value over the synthesis period of the analysed biomass [23,25,29,39]. Nevertheless, their representativeness sometimes seemed to be questionable [10,40].

The present study analyses the relationship between $\delta^{13}$C and WUE$_i$ values of each genotype and evaluates whether each parameter was representative of the WUE. The values of both parameters were in the same range as those obtained in different previous research [10,12,13,29,36].

The relationship between the conductance and logarithm of WUE$_i$ (Figure 1) followed the expected distribution according to Tortosa et al. [36]. The genotypes analysed in 2015 had moderate ($g_s < 0.15$) and severe water stress ($g_s < 0.05$) caused by low precipitation (Table 1) in the summer months [41]. This fact causes a high WUE$_i$ value due to the strong association between $g_s$ and WUE$_i$ [8,33]. The high variability between the strains and fields led to a high correlation ($R^2 = 0.77$) between the values.

The results obtained between clones have the same ranges of values as those obtained in the work of Buesa et al. [13] and Tortosa et al. [36]. As expected, the same high and low WUE genotypes were identified in both parameters.

This study is pioneering in carrying out a detailed comparison of the two parameters for measuring the discriminative ability of the WUE. A good relationship (Pearson correlation = 0.719) between $\delta^{13}$C and WUE$_i$ was found (Figure 2) in consensus with previous works [23,42,43]. The represented points did not fit the line more closely due to the dif-

ferences between the different conditions of the experimental years and the differences in field management. The wide range in water status ($g_s$: <0.05 to >0.5) was reflected in a large range of $\delta^{13}C$ values ($-20.7‰$ to $-29.7‰$). Differences between cultivars in the same plot (Garnacha and T-rootstock) and intra-cultivar in other plots (Tempranillo in Roda and La Grajera) were greater in $\delta^{13}C$ values. The Garnacha genotypes (2018) did not show the relation between parameters, obtaining a cloud of points with less variation in the $WUE_i$ range (60–80 mmol $CO_2$ mol$^{-1}$ $H_2O$) than in $\delta^{13}C$ values (from $-27‰$ to $-23.5‰$). $WUE_i$ does not always reflect the WUE of the whole plant [10,44].

The residual rate of $WUE_i$ tends to have a higher error (Figure 3) due to the high variability mentioned [44]. This fact may be due to the unique characteristics of each sampled plant within the crop [45,46], which results in a bigger oscillation in $WUE_i$ values. For this reason, a single sampling of leaf gas exchange may be insufficient to define the WUE of the plant.

In this study, the $\delta^{13}C$ data showed differences in WUE between plots and cultivars that had not been observed in $WUE_i$ (Figure 2 and Tables 3 and 4). $WUE_i$ data were representative of the time of measurement and, therefore, of the environmental conditions and plant water status at the measurement time. In contrast, $^{13}C$ isotopic discrimination was an accumulative parameter over the time of formation of the dry mass analysed [39,47]. For this reason, the analyses of the berries in the ripening phase (synthesis and accumulation of sugars) resulted in a good estimation of WUE because they reflected the plant water deficit during the driest period [23,48].

$WUE_i$ and $\delta^{13}C$ methods can discriminate between the best and worse genotypes in WUE (Figure 5) [13]. Nevertheless, $^{13}C$ discrimination analyses showed a better resolution for WUE clone identification, which could be related to a wider range of its values among the efficient and inefficient genotypes. Consequently, this parameter allowed a better clone identification of WUE in breeding programs.

Moreover, the choice of analysing grape samples for $\delta^{13}C$ could be an interesting alternative for clone selection programmes because it reflects the average economy of the water use during the synthesis process of the dry mass analysed [13,23]. Furthermore, $^{13}C$ isotope discrimination analysis is much easier to carry out under field conditions, because it only required a representative sampling of berries and provided an integrated WUE value over 1–2 months of berry filling that coincided with the water stress moment [24,49].

Moreover, the measurement of $WUE_i$ has some technical inconveniences: specialised instrumentation, a longer time-consuming measurement which was limited to certain hours of the day, dependent on environmental conditions [36], and large differences rates in the plant [44–46]. These reasons make data collection somewhat difficult and limit the number of samples that could be collected in a day. In contrast, the $\delta^{13}C$ parameter shows the advantage that the number of collected samples can be much higher. Moreover, the sample collection was easier and was not dependent on environmental conditions.

## 5. Conclusions

The analytical methodology used allowed a fair evaluation of WUE in 70 genotypes of two cultivars (Tempranillo and Garnacha) and a collection of T-rootstock clones. The field experiment was based on $^{13}C$ discrimination and leaf gas exchange ($WUE_i$) values. Clones with high and low WUE were defined based on both parameters. Furthermore, a good correlation between the two parameters was obtained, indicating that both parameters were good indicators to define WUE.

This work provides results to estimate that carbon isotopic discrimination was a more interesting parameter than $WUE_i$. The main reason was that the $\delta^{13}C$ found differences in the groups that the $WUE_i$ values could not. In addition, this parameter had a high resolution defining WUE among clones of the same group and between different groups.

In addition, the analysis of $^{13}C$ berry samples offers other technical advantages such as the possibility to collect a larger number of samples in one day, which can be stored for later measurements, avoiding the problems derived from the instantaneous measurement,

the independence of environmental and day conditions, and the integration of the WUE over the berry filling period.

**Author Contributions:** Data curation, I.T., C.D., A.P. and A.M.; formal analysis, I.T., C.D.; funding acquisition, J.M.E. and H.M.; methodology, J.M.E. and H.M.; project administration, J.M.E.; writing—original draft, A.M. and H.M.; writing—review and editing, A.M., H.M. and A.P. All authors have read and agreed to the published version of the manuscript.

**Funding:** This work was carried out with financial support from the Spanish Ministry of Science and Technology (FEDER/Ministerio de Ciencia, Innovación y Universidades–Agencia Estatal de Investigación/_AGL2017-83738-C3-1-R) and a pre-doctoral fellowship (PRE2019-089110) with a narrow collaboration inside the Associated Unit ICVV-INAGEA.UIB.

**Data Availability Statement:** Not applicable.

**Acknowledgments:** The authors would like to thank M. Ribas-Carbó and collaborators of the UIB for their support in $\delta^{13}C$ measurements. We also want to thank the collaboration of the Instituto de las Ciencias de la Vid y el Vino (ICVV) and Viveros Provedo S.A, Bodegas Roda S.A. y Vitis Navarra Seleccion S.I. for providing us the plant material.

**Conflicts of Interest:** The authors declare no conflict of interest. The funders had no role in the design of the study; in the collection, analyses, or interpretation of data; in the writing of the manuscript, or in the decision to publish the results.

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
