# Peer review of "Comparing Selection Criteria to Select Grapevine Clones by Water Use Efficiency"

_agronomy, doi:10.3390/agronomy12081963_

Round 1
Reviewer 1 Report
The purpose of the article “Comparing selection criteria to select grapevine clones by water use efficiency” is to report a valuable the variability of WUE between clones of the Garnacha and Tempranillo cultivar and Tempranillo on different rootstocks and evaluate the discrimination capacity of gas exchange and 13C isotopic ratio in two years of field growing vines data. The authors present differences in WUE found between genotypes in different cultivars (Tempranillo and Garnacha) and T-rootstock clones in different plots and years. Also demonstrated that δ13C is a very interesting parameter for a clonal selection based on WUE due to its capacity to integrate the water efficiency of the period in which the plant was usually more exposed to the water deficit. The article, however, must be improved in terms of writing since some grammar and syntax errors are present in the manuscript. They should address the subject and critically review the information from the literature.
My suggestions:
The authors need to revise the title of the paper in a more meaningful way. (Selection criteria to select grapevine clones by water use efficiency);
The abstract is written in a way lacks logic. It should highlight the salient findings more critically;
Keywords are present in the title, choose others (Grapevine);
Introduction need more convincing rational for this article. In the introduction, there was a lack of talking about the main criteria used in the genetic breeding of grapes, as well as presenting new perspectives on this topic;
The introduction has long paragraphs, I suggest reducing the size of the paragraphs.
Provide experimental work plan at the start of M&M. No detail description is available about the experiment.
For IRGA analysis has an auxiliary light source been coupled? Generally, the coupling of a light source with 995 µmol m-2-1 is used for evaluations.
Authors should discuss the results integrally. The discussion is based on individual results. I suggest that integrating the results will give more value to the work. I suggest that you discuss by integrating all your results. You can use principal component analysis.
The results of this study are not fully explained therefore the interpretation of the results is very difficult. The author needs to provide the % increase or decrease rather than just writing ''significantly increased….''.
Table 2, 3, and 4: Please provide standard deviation of the results?
The discussion is poorly written hence, needs rewriting. The discussion should be further strengthened by adding some more relevant papers. The literature search is insufficient, only few related research papers in the past three years are cited, add the latest research results appropriately.
Rewrite the conclusion! It needs to be much improved.
Author Response
Dear Editor,
Thank you very much for allowing us to revise and resubmit this manuscript for publication in Agronomy. We thank and appreciate the careful and constructive review of the referees. Their suggestions were important for the improvement of the manuscript. We took into consideration most of their suggestions, and we hope the new version of the manuscript will meet your expectations. Details of point-by-point responses to the reviewer ́s comments in the revised manuscript are included in the revision notes folder.
We hope that the revised manuscript is now suitable for publication in Agronomy.
With best regards,
Andreu Mairata Pons
Instituto de Ciencias de la Vid y del Vino (CSIC, Universidad de La Rioja, Gobierno de La Rioja).
Finca La Grajera, Carretera de Burgos km 6, 26007 Logroño, Spain.
E-mail: andreumairata112@gmail.com
Reviewer 1:
1# The authors need to revise the title of the paper in a more meaningful way. (Selection criteria to select grapevine clones by water use efficiency)
We are thankful to the reviewer for the insightful suggestion, which is certainly important to improve the manuscript. We have considered changing the title but we have not found one that we like better to contextualise the subject and the objectives of the experiment.
2# The abstract is written in a way lacks logic. It should highlight the salient findings more critically.
Thank you for your opinion. The abstract has been rewritten.
3# Keywords are present in the title, choose others (Grapevine).
Thank you. The keywords have been revised.
4# Introduction needs a more convincing rationale for this article. In the introduction, there was a lack of talking about the main criteria used in the genetic breeding of grapes, as well as presenting new perspectives on this topic;
A paragraph about genetic breeding grapes has been added (line 58). The introduction has been rethought by integrating all the concepts and trying to contextualise the objectives and works of the study.
5# The introduction has long paragraphs, I suggest reducing the size of the paragraphs.
In general, paragraphs are not very long (maximum 8 lines). Even so, an attempt has been made to compress the concepts as much as possible. The ideas have also been reorganised and the size of some paragraphs has been reduced. Now It is very difficult to reduce paragraphs without reducing information.
6# Provide an experimental work plan at the start of M&M. No detailed description is available about the experiment.
An experimental work plan (line 110) was added in paragraph 2.1 to better explain the procedure of the experiment.
7# For IRGA analysis has an auxiliary light source been coupled? Generally, the coupling of a light source with 995 µmol m-2-1 is used for evaluations.
As mentioned in line 129, no external light has been used. It has always been measured at PAR intensities above 1500 µmol m2 s-1, preventing the lack of light from limiting or altering the photosynthesis values.
8# Authors should discuss the results integrally. The discussion is based on individual results. I suggest that integrating the results will give more value to the work. I suggest that you discuss this by integrating all your results. You can use principal component analysis.
Thank you for your comment. The discussion has been changed trying to discuss the most relevant points of the paper in a more joined-up way. In our experiment, we believe that making a scatter plot or PCA would not be very convenient because we have studied mainly two parameters and, in this kind of graph, it would be very poor.
9# The results of this study are not fully explained therefore the interpretation of the results is very difficult. The author needs to provide the % increase or decrease rather than just writing ''significantly increased….''.
Thank you for your suggestion. Changes have been made to be more accurate in reporting the results.
10# Table 2, 3, and 4: Please provide the standard deviation of the results.
In tables 2, 3 and 4 the standard deviation of all values is given.
11# The discussion is poorly written hence, needs rewriting. The discussion should be further strengthened by adding some more relevant papers. The literature search is insufficient, only a few related research papers in the past three years are cited, add the latest research results appropriately.
The discussion has been improved. It has been complemented by other work (9 references more). The work has been searched for papers working with the parameters used to measure WUE. Nevertheless, there are not many references on clonal selection by WUE, and even less with grapevine.
12# Rewrite the conclusion! It needs to be much improved.
The conclusions have been rewritten in an attempt to capture the main ideas of the study and its future impact.

Reviewer 2 Report
Comment 1#. In the Introduction section, line 34, you say that "vineyards are a traditional Mediterranean crop with a long history that completes its biological cycle from April to October". First of all, I think would be more precise to use grapevine and not vineyards,, since you qre speaking about plant lifecycle. Second, I would suggest to reformulate the phrase less strictly, because depending by the vintage, geographical area and cultivar, the span of the biological cycle can vary.
Comment 2#. In the Materials and Methods section, please clarify how many biological replicates did you use. It is not clear. Moreover, I ask to the authors why they did not decide for two consecutive years insteade of performing the study in 2 distant years.
Comment 3#. Paragraph 2.2: Why did you use only the intrinsic WUE and not also whole plant level WUE? I believe it could have been another usefull parameter for screening evaluation, as well as dry biomass production. Moreover, if possible add mesophillar conductance (gm) results, because in the literature is reported that WUEi can be disturbed by it.
Comment 4#. Paragraph 2.3: I was wandering why you did not analyse oxygen stable isotopes too: when used together with carbon isotopes, they can be usefull too establish sources of WUE variation.
Comment 5#. Paragraph 2.4: please, state which test did you use to verify data distribution and homoscedasticity, since you used ANOVA, a parametric analysis. Moreover, I suggest to authors to clarify which experimentzl design they used: split plot? split split plot? completely randomized? Please clarify.
Comment 6#. I suggest authors to reformulate the conclusion chapter in a slightly more elaborate and convincing way.
Author Response
Dear Editor,
Thank you very much for allowing us to revise and resubmit this manuscript for publication in Agronomy. We thank and appreciate the careful and constructive review of the referees. Their suggestions were important for the improvement of the manuscript. We took into consideration most of their suggestions, and we hope the new version of the manuscript will meet your expectations. Details of point-by-point responses to the reviewer ́s comments in the revised manuscript are included in the revision notes folder.
We hope that the revised manuscript is now suitable for publication in Agronomy.
With best regards,
Andreu Mairata Pons
Instituto de Ciencias de la Vid y del Vino (CSIC, Universidad de La Rioja, Gobierno de La Rioja).
Finca La Grajera, Carretera de Burgos km 6, 26007 Logroño, Spain.
E-mail: andreumairata112@gmail.com
Reviewer 2:
Comment 1#. In the Introduction section, line 34, you say that "vineyards are a traditional Mediterranean crop with a long history that completes its biological cycle from April to October". First of all, I think would be more precise to use grapevines and not vineyards, since you are speaking about plant lifecycle. Second, I would suggest reformulating the phrase less strictly, because depending on the vintage, geographical area and cultivar, the span of the biological cycle can vary.
We are thankful to the reviewer for the insightful suggestions, which are certainly important to improve the manuscript. As you can see in the paragraph beginning on line 37, the introduction of the issue has been reviewed and tried to be more general.
Comment 2#. In the Materials and Methods section, please clarify how many biological replicates did you use. It is not clear. Moreover, I ask the authors why they did not decide for two consecutive years instead of performing the study in 2 distant years.
As mentioned in lines 127 and 135, both parameters (WUEi and 13C) have been measured on the same 4-6 plants of each clone. Even so, it has been specified again in section 2.3 (135).
Regarding the years of measurement (2015 and 2018), two years have been left without measurements due to the impossibility of sample collection. This is because, among other circumstances, the research team is located in another community in Spain (Balearic Islands) and it was impossible to collect data for both parameters these years (2016 and 2017).
Comment 3#. Paragraph 2.2: Why did you use only the intrinsic WUE and not also whole plant level WUE? I believe it could have been another usefull parameter for screening evaluation, as well as dry biomass production. Moreover, if possible add mesophillar conductance (gm) results, because in the literature is reported that WUEi can be disturbed by it.
Thank you for your observation. Certainly, analysing the WUE of the whole plant would have been very interesting. However, this paper focuses on analysing the differences and the ability to characterise the WUE of two parameters (WUEi and 13C). In addition, pruning weight and yield/irrigation are influenced by crop management and environmental conditions in the early stages of growth, when the plant is not under water stress. In this sense, mesophyll conductance (gm) affects photosynthesis and WUEi, analysing it and discussing the results is neither the purpose nor the interest of this work because it would not provide interesting information for the proposed objectives.
Comment 4#. Paragraph 2.3: I was wondering why you did not analyse oxygen stable isotopes too: when used together with carbon isotopes, they can be useful to establish sources of WUE variation.
In this experiment, the analysis of the oxygen isotope has several drawbacks due to the lack of data on the water applied (irrigation) and the absence of samples of irrigation water from all the plots. Furthermore, it is understood that the oxygen isotope is well related to stomatal conductance. In contrast, the carbon isotope is related to WUEi (integration of photosynthesis and stomatal conductance), which is more complete and more comparable with WUEi measurements.
On the other hand, only two studies have been found where the oxygen isotope was used in clonal selection in pine and eucalyptus (Xu et al. 2000; Dvorak et al. 2012). No work has been found in which the oxygen isotope was used in clonal selection in grapevine. In addition, the work of Gomez-Alonso et al. (2020) proposed that carbon isotopic discrimination was a better estimator of WUE in berries because it found more differences between cultivars than with the 18O. Although it would be interesting to see the discriminatory capacities of this isotope in grapevine breeding programmes in future experiments.
- Dvorak, W. S. (2012). Water use in plantations of eucalypts and pines: a discussion paper from a tree breeding perspective. International Forestry Review, 14(1), 110-119.
- Xu, Z. H., Saffigna, P. G., Farquhar, G. D., Simpson, J. A., Haines, R. J., Walker, S., ... & Guinto, D. (2000). Carbon isotope discrimination and oxygen isotope composition in clones of the F1 hybrid between slash pine and Caribbean pine in relation to tree growth, water-use efficiency and foliar nutrient concentration. Tree Physiology, 20(18), 1209-1217.
- Gómez‐Alonso, S., & García‐Romero, E. (2010). Effect of irrigation and variety on oxygen (δ18O) and carbon (δ13C) stable isotope composition of grapes cultivated in a warm climate. Australian Journal of Grape and Wine Research, 16(2), 283-289.
Comment 5#. Paragraph 2.4: please, state which test did you use to verify data distribution and homoscedasticity, since you used ANOVA, a parametric analysis. Moreover, I suggest to the authors clarify which experimental design they used: split plot? split split plot? completely randomized? Please clarify.
It is an interesting observation. Now, in line 150 you can see more details about the data analysis. Moreover, the experimental design was defined in paragraph 2.1 (line 109).
Comment 6#. I suggest authors reformulate the conclusion chapter in a slightly more elaborate and convincing way.
Thank you for your contribution. The conclusions have been rewritten in an attempt to capture the main ideas of the study and its future impact.
